# Modified secreted alkaline phosphatase as an improved reporter protein for *N*-glycosylation analysis

**Mariusz Olczak** \*, **Bożena Szulc**

Laboratory of Biochemistry, Faculty of Biotechnology, University of Wrocław, Wrocław, Poland

\* mariusz.olczak@uwr.edu.pl

## Abstract

*N*-glycosylation is a common posttranslational modification of proteins in eukaryotic cells. The modification is often analyzed in cells which are able to produce extracellular, glycosylated proteins. Here we report an improved method of the use of genetically modified, secreted alkaline phosphatase (SEAP) as a reporter glycoprotein which may be used for glycoanalysis. Additional *N*-glycosylation sites introduced by site-directed mutagenesis significantly increased secretion of the protein. An improved purification protocol of recombinant SEAP from serum or serum-free media is also proposed. The method enables fast and efficient separation of reporter glycoprotein from a relatively small amount of medium (0.5–10 ml) with a high recovery level. As a result, purified SEAP was ready for enzymatic de-glycosylation without buffer exchange, sample volume reductions or other procedures, which are usually time-consuming and may cause partial loss of the reporter glycoprotein.

## 1. Introduction

In the majority of eukaryotic cells, glycosylation is among the most frequent posttranslational modifications of macromolecules, i.e., glycoproteins, proteoglycans, and lipids. Cellular glyco-conjugates play a variety of fundamental roles in the growth and development of eukaryotes, as well as in the cell surface recognition of hosts by pathogens. Therefore advanced knowledge of glycosylation mechanisms is of crucial importance. The glycan moiety is synthesized with the involvement of glycosyltransferases. Protein glycosylation is a very complex process that employs almost 200 glycotransferases, enzymes that determine which proteins are to became glycoproteins, the positions of glycans on those proteins and the glycan structures assembled [1]. Among different types of protein posttranslational modifications, *N*-glycosylation is one of the most common. This modification is dependent on specific three-amino-acid motifs in glycoproteins (sequons). The sequon for *N*-glycosylation is either N-X-S or N-X-T, where X is any amino acid except proline. In some cells, especially those with high activity of secretory pathways (ER-Golgi processing), most proteins are subjected to *N*-glycosylation [2]. In eukaryotic tissues most *N*-glycans are converted from initial, high mannose type to mature forms, containing additional *N*-acetylglucosamine, galactose, fucose and sialic acid residues. However, a detailed analysis of *N*-glycans covalently attached to glycoproteins of mammalian cell

**Data Availability Statement:** All relevant data are within the manuscript and its Supporting Information files.

**Funding:** This research was supported by funding from National Science Center (NCN), Kraków,

Poland, Website: www.ncn.gov.pl, Grant No. 2016/21/B/NZ5/00144 (MO) The funders had no role in study design, data collection and analysis, decision to publish, or preparation of the manuscript.

**Competing interests:** The authors have declared that no competing interests exist.

lines is not an easy task, not only because of high heterogeneity of glycan structures. Detailed characterization of them may be performed by investigating fragmented glycopeptides using a sophisticated, high-throughput mass spectrometry approach; however, most research may prefer analyses of free glycans, released enzymatically or chemically from glycoproteins. Both strategies use total protein extracts or glycoprotein fractions isolated from biological material, usually eukaryotic cell lines. Classical methods of glycoprotein isolation procedures include detergent extraction of the total protein pool, followed by time-consuming purification steps, usually with relatively low recovery rates. Additionally, in many frequently used mammalian cell lines, the most common problem in such direct methods of isolation is a large amount of high-mannose type, not-mature glycans, abundant in ER or early Golgi compartments [3, 4]. Those structures are useless for investigation of the majority of glycotransferase activity in regard to elucidating the function and regulation of the very complex glycotransferase system, a phenomenon still not completely understood in mammals. The intracellular *N*-glycan pool may also contain other non-mature intermediates, which might produce a unsatisfactory "background", visible in *N*-glycan profiles. Recently, to overcome the problems listed above, new methods based mostly on removing high-mannose structures have been proposed. However, these are still time-consuming and relatively complicated procedures [5]. The strategy is based on sequential application of specific endoglycosidases (Endo H to remove high mannose glycans and then PNGase F to release oligosaccharides); however, it seems that every additional step makes such an approach less efficient and may be questionable, especially for quantitative analyses of multiple samples. The enrichment of mature *N*-glycans was also reported with the use of a C18 column to remove high-mannose structures [6]. However, this is still quite a laborious procedure. Direct trypsin digestion of glycoproteins exposed on the cell membrane was also proposed [4]. Such an approach seems to be promising for analyses of mature *N*-glycans, which undergo the full process of biosynthesis and maturation in ER and Golgi, and are exposed on the cell surface; however, in such a scenario, there is always a danger of contamination by some endogenous serum proteins, attached to the surface of the cells, especially if they are grown in serum-containing media, typically used for propagation of most mammalian cell lines. Such a background may distort the real glycosylation profile, making the results unreliable [7, 8].

To solve all these problems, instead of working with total cell lysates, many researchers use reporter, secreted glycoproteins in genetically modified cells. A few secretable glycoproteins, such as erythropoietin (EPO), human granulocyte macrophage colony stimulating factor (hGM-CSF), interferon and SEAP [9–14], have been tested. This approach has some unquestionable advantages. Firstly, only mature glycoproteins (and its *N*-glycans) are analyzed. Secondly, the reporter protein may be quite easily isolated, preferably by simple chromatographic methods; there is no need to purify the reporter agent from very complex cell lysates, which contain not only hundreds of proteins but also non-protein material, such as lipids, nucleic acids and other high- and low-molecular weight molecules. Finally, changes in glycosylation may be analyzed in detail, in contrast to examination of cell lysate *N*-glycan pools, where tiny perturbations in glycosylation may be not visible, because of the high complexity of such cellular *N*-glycan profiles.

One of the commonly used reporter glycoproteins for analysis of glycosylation is secreted alkaline phosphatase (SEAP), known for its strong secretion signal and used many times in the past [9, 10]. The secreted protein is derived from original human placental alkaline phosphatase, by removing the GPI anchor motif [15]. However, the use of native SEAP as a reporter glycoprotein for glycoanalysis also appears not to be a perfect choice. Firstly, the protein, secreted to the media, must be purified, usually with a few steps of combined chromatographic and other laboratory techniques. To avoid many laborious and time-consuming procedures

during purification, serum-free media in tissue cultures may be introduced; however, this is not possible for many cell lines. Secondly, the native SEAP is not highly glycosylated, with only one occupied *N*-glycosylation site at position 272. This means that a relatively large amount of purified SEAP is required to perform satisfactory experiments.

The aim of this study was to introduce a new proteomic tool, useful in analyses of glycosylation machinery in a broad spectrum of mammalian cell lines. Our goal was to develop a new genetic vector encoding a reporter glycoprotein, able to be purified faster, more efficiently, and to be easily monitored during secretion and isolation. Our plan also expected new glycosylation sites to be introduced in SEAP, to increase the amount of *N*-glycans released from purified, homogeneous enzyme, which may be used in future structural studies. We were also hoping for more efficient secretion of the modified protein, a phenomenon observed for other proteins with extended glycosylation [16].

Here, we describe an improved version of a secreted reporter protein system, based on the psiTEST plasmid from Invivogen, which codes for the sequence of the SEAP. The main goal of this project was to design a new version of the glycoprotein which: 1) may be synthesized and secreted to the culture media by transiently transfected mammalian cells of different types at higher rates than previously reported; 2) the secretion level may be monitored in seconds or minutes using simple colorimetric assay; 3) the purification procedure is fast, efficient, and is also not dependent on type of serum and culture media; 4) the reporter protein possesses at least three functional glycosylation sites, in contrast to the original SEAP, which contains only one functional *N*-glycosylation site [17, 18].

To achieve all these requirements, we constructed the modified genetic vector (derived from a commercially available psiTEST plasmid), coding for changed SEAP. We tested GST, HA or 6×His tags attached at the C-terminus of SEAP for efficient purification. We also introduced new glycosylation sites. From 29 analyzed constructs, 2 were chosen as the best in view of secretion level and *N*-glycosylation pattern. The finally selected constructs were tested on HEK293TT, HepG2 and CHO cells.

## 2. Materials and methods

### 2.1. Cloning and site-directed mutagenesis

As a starting point, the psiTEST plasmid (Invivogen) was utilized. *SEAP* is an optimized alkaline phosphatase gene engineered to be secreted. The enzyme catalyzes the hydrolysis of QUANTI-Blue phosphatase substrate (Invivogen) producing a purple to dark blue end product that can be directly analyzed in seconds by the naked eye or read spectrophotometrically at 620–655 nm. SEAP inserts containing additional 6×His, HA and GST tags were amplified by PCR performed with Q5 thermostable DNA polymerase (New England Biolabs, NEB) according to the NEB standard protocol. For 6×His and HA constructs, the psiTEST vector was used as a template. In the case of the GST construct, the plasmid PSF-OXB20-GST was utilized as a template for PCR amplification of the SEAP-GST insert (see S1 Table and S1 and S2 Figs). Q5 site-directed Mutagenesis Kit (NEB) was used to introduce new *N*-glycosylation sites in SEAP. Two strategies were employed to create new sequons: direct change of an amino acid residue located upstream (at position -2, mutations #101, #278, #477, #493) to existing threonine or serine to asparagine, or replacing an amino acid residue placed downstream (at position +2, mutations #38, #109, #152) of existing asparagine to serine or threonine.

### 2.2. Cell culture and transfection

HEK293TT, HepG2 and CHO cells were grown under humidified atmosphere (at 37˚C in 5% $CO_2$) in Dulbecco's Modified Eagle's Medium–high glucose for HEK209T and HepG2 cells

(DMEM, Merck), and Minimum Essential Medium Eagle with alpha-modification for CHO cells (α-MEM, Merck). Medium was supplemented with 10% fetal bovine serum (FBS, Biowest), 100 U/ml penicillin, and 100 mg/ml streptomycin (Merck). For transfection, cells were growth in 6-well plates or 10-cm dishes to 50–60% confluence. Cells were transiently transfected with plasmids using the FuGENE 6 transfection reagent (Promega) at the transfection reagent:DNA ratio 3:1. Media were changed 24 hours after transfection and cells were grown for the next two days in respective serum-containing medium or Opti-MEM serum-free medium (Thermo Fisher). Then, media were collected and centrifuged at 500×*g* for 5 minutes, and supernatants were collected for future analysis.

## 2.3. Enzymatic assays

Media containing SEAP were collected and examined with QUANTI-Blue phosphatase substrate (Invivogen), according to the producer's manual. Briefly, for preliminary screening, 2-20 µl of culture medium was mixed with 180 µl of fresh QUANTI-Blue reagent in low-protein binding 96-well plates. The purple color was developed at 37˚C for 30 seconds to 10 minutes, with gentle shaking. If precise measurements were preferred, absorbance of the reaction mixtures was analyzed at 600 nm in a standard 96-well microplate reader.

Activity assays for purified variants of SEAP were performed in 0.1 M Tris/HCl buffer pH 9.5, supplemented with 5 mM $MgCl_2$, 0.1 M NaCl and 0.2% Triton X-100, with 5 mM *p*-nitrophenylphosphate (pNPP, Sigma-Aldrich) as a substrate. The final volume of the reaction mixture was 0.5 ml. The reactions were performed at 30˚C for 5 minutes and stopped by addition of 1.5 ml of 0.5 M NaOH. The liberated *p*-nitrophenol was determined by measuring the absorbance at 405 nm using a Beckman DU-640 spectrophotometer and the calibration curve of *p*-nitrophenol (pNP, Sigma-Aldrich) prepared under the same conditions. One enzymatic unit (U) was described as the amount of the enzyme that hydrolyzes 1 µmol of pNPP per minute at the test conditions described above. Specific activity of the enzyme was defined in units calculated per mg of protein (U/mg). All enzymatic reactions and enzyme dilutions were performed in low-protein binding 1.5 ml Eppendorf tubes.

## 2.4. Purification of SEAP from culture media

1 to 10 ml of the medium expressing phosphatase activity was used for isolation of the homogeneous enzyme. 10× concentrated solution of PBS was added to the medium to the final concentration of 40 mM phosphate buffer, pH 7.4, containing 0.3 M sodium chloride. The medium was also additionally supplemented with imidazole and Triton X-100 to reach 5 mM and 0.5%, respectively. Washing buffer (2×PBS with 0.5% Triton X-100 and 5 mM imidazole) was used for initial conditioning of magnetic beads and for all washings. Magnetic beads (Ni-NTA Magnetic Agarose Beads, Jena Bioscience) were added to the supplemented medium (5 µl of 25% conditioned beads suspension per 1 ml of the SEAP medium) and incubated at room temperature for 15 to 120 minutes on a rotary shaker in slow motion mode (approximately 10 rotations per minute). After the binding step, beads were separated on a magnetic rack and washed 4 times, each time with 0.8 ml of washing buffer. Elution was performed with 40 µl of 1% SDS, containing 80 mM DTT (2× concentrated Glycoprotein Denaturing Buffer, New England Biolabs) at 100˚C for 15 minutes with shaking at 400 rpm in a dry block equipped with a hot (105˚C) lid, to avoid evaporation. After cooling, supernatant was separated from beads using a magnetic rack. Alternatively, 250 mM imidazole incubation (RT, 15 minutes, with shaking) or 2×Laemli sample buffer (100˚C, 5 minutes) was used for elution. Imidazole elution was applied for isolation of active SEAP to perform enzymatic analyses of phosphatase variants.

In the case of purification using the Ni-NTA column, the same buffers were used for sample preparation and washing steps. Ni-NTA 150 columns (Macherey-Nagel) were equilibrated with washing buffer, samples were loaded on the column, then 5 ml of wash buffer was applied to remove impurities and weakly bound material. Elution was performed twice using 250 μl of 20 mM Tris/HCl buffer, pH 8.0, containing 250 mM imidazole.

A similar purification procedure as in the case of the Ni-NTA magnetic bead experiments was used to purify SEAP-HA protein using anti-HA magnetic agarose beads (Bimake) and 2× Glycoprotein Denaturing Buffer or 2× Laemmli buffer as an elution agent. In this procedure, imidazole was not present in washing and elution mixtures.

## 2.5. SDS-PAGE

SDS-polyacrylamide gel electrophoresis was performed on 10% gels. Gels were stained with Colloidal Coomassie Blue (PageBlue Staining Solution, Fermentas).

## 2.6. Protein estimation

Concentration of purified and denatured SEAP was performed using a modified Bradford method [19] with Roti-Nanoquant reagent (Carl Roth). The SEAP concentration of non-denatured protein utilized for enzymatic analyses was determined from its molar coefficient, calculated from the predicted molecular mass (54.97 kDa) and aromatic amino acid content of the mature SEAP. The molar coefficient measured at 280 nm was estimated to 47 580 ($A_{280}$ $M^{-1}cm^{-1}$), which is equivalent to $A_{280} = 0.866$ for 0.1% concentration of the homogeneous enzyme. The signal peptide cleavage site (S1 Fig) reported by the psiTEST plasmid vendor (Invivogen) was additionally confirmed using SignalP-5.0 software [20].

## 2.7. *N*-glycan profiling

Purified SEAP obtained after elution with 2× Glycoprotein Denaturing Buffer was diluted twice with Milli-Q water. Deglycosylation was performed using recombinant PNGase F, according to the recommended New England Biolabs protocol. Briefly, diluted SEAP samples were supplemented with Nonidet NP-40 and phosphate buffer, pH 7.5, to the final concentration of 1% and 50 mM, respectively, with the recommended PNGase concentration (500 U per reaction). The standard time of deglycosylation was 16 hours; however, shorter incubations (30 minutes to 2 hours) were also tested, without a significant decrease in deglycosylation rate. Released *N*-glycans were isolated and fluorescently labeled on the non-reducing end with 2-aminobenzamide (2-AB) as previously described [21]. Briefly, the deglycosylation mixtures were applied to SuperClean EnviCarb SPE columns (Supelco), previously primed with 3 ml of acetonitrile/0.1% TFA and equilibrated with 6 ml of Milli-Q water. After sample loading, columns were washed with 3 ml of water, followed by 6 ml of 3% acetonitrile containing 0.1% TFA. Glycans were eluted using 2 ml of 50% of acetonitrile containing 0.1% of TFA, dried under vacuum and labelled with 2-aminobenzamide. The labelling mixture was prepared by dissolving 6.3 mg of 2-aminobeznzamide (Sigma-Aldrich) in 0.1 ml of DMSO/acetic acid 3.5/1.5 (v/v) mixture. Then the solution was transferred to a clean tube containing 7.8 mg of sodium cyanoborohydride (Sigma-Aldrich). 5 μl of this solution was used for labelling of each *N*-glycan sample in a dry heating block, for 3 hours at 65°C. Labelled glycans were separated from the excess fluorescent stain on 10 mm diameter discs made of 1 mm thick blotting paper (Whatman 3M), and placed on a glass support. Paper discs were previously equilibrated with 5 ml of 30% acetic acid followed by 1 ml of acetonitrile. The total volume of labelling mixture (5 μl) was spread on the surface of the paper and left for 15 minutes at RT. Then, paper discs

were washed with 1 ml of 100% acetonitrile followed by 6 × 1 ml of 96% acetonitrile. 2-AB labelled glycans were eluted using 3 x 0.5 ml of Milli-Q water and dried under vacuum.

Purified N-glycans were separated on a normal-phase GlycoSepN amide column (Pro-Zyme) in high-salt gradient mode, as previously reported [21]. Signals were recorded using the Perkin Elmer series 200 HPLC system, equipped with a Hitachi fluorescence detector (330 nm excitation, 420 nm emission). To remove high-mannose structures from labeled glycan mixtures, reactions with 1–2,3,6 α-mannosidase in Glycobuffer 1 supplemented with Zn ions were performed according to the producer's manual (NEB). After separation on the GlycoSepN column, the quantitative analysis used peak area estimation software (TotalChrom, Perkin Elmer) to analyze the relative amount of high-mannose and complex *N*-glycan pools. The relative amount of the high-mannose (non-mature) structures was calculated as the ratio of the peak area of the final mannosidase product (shown in Fig 4), derived from all high-mannose structures, divided per total area of the same peak and areas of all other structures which were resistant to mannosidase digestion.

### 2.8. Statistical analysis

One-way ANOVA test and Dunnett's post-hoc test were employed to analyze statistical significance of the results. All statistical analyses were performed using GraphPad Prism software.

## 3. Results and discussion

### 3.1. Analysis of phosphatase activity of recombinant SEAP with tags attached at the C-terminus

As a starting point, we decided to use the commercially available psiTEST plasmid, which codes for the sequence of secreted alkaline phosphatase. The vector was originally designed by Invivogen to test the efficiency of the RNAi strategy by attaching a sequence of a gene of interest at the 3' terminus of the SEAP reporter sequence. The same vendor also offers a very sensitive QUANTI-Blue chromogenic substrate, which may detect phosphatase activity directly in mammalian cell culture media. The commonly used HA, 6×His and GST fusion tags introduced to facilitate purification procedures were tested. HEK293T cells were transiently transfected with HA, 6×His and GST vectors in the presence of FuGENE transfection reagent and media were analyzed with phosphatase QUANTI-Blue substrate. As shown in Fig 1, SEAP-HA and SEAP-6×His were almost equally efficiently secreted, in contrast to SEAP-GST, which exhibits only traces of enzymatic activity in the culture medium. It seems that the relatively large size of GST (25 kDa) would be a negative factor for secretion of the enzyme. Because of that, for the next experiments we chose only the first two vectors.

### 3.2. Testing for optimal conditions for isolation of homogeneous SEAP from culture media

In the preliminary experiments, we found that the secretion rates for cells propagated in media supplemented with 10% serum and in serum-free medium were almost identical. However, as a starting point, we decided to test the SEAP isolation from serum-containing media only. If this would work, the procedure should be suitable also when serum-free medium is used. At the beginning, for isolation of 6×His tagged protein, Ni-NTA magnetic beads, Ni-NTA columns and the Ni-NTA agarose batch procedure were tested. Although it is impossible to perform any reliable statistics due to many multi-step procedures tested in dissimilar conditions, from those methods of purification only magnetic beads bound SEAP-6×His protein with acceptable recovery (usually between 85% and 95%, never below 75%). We were surprised to

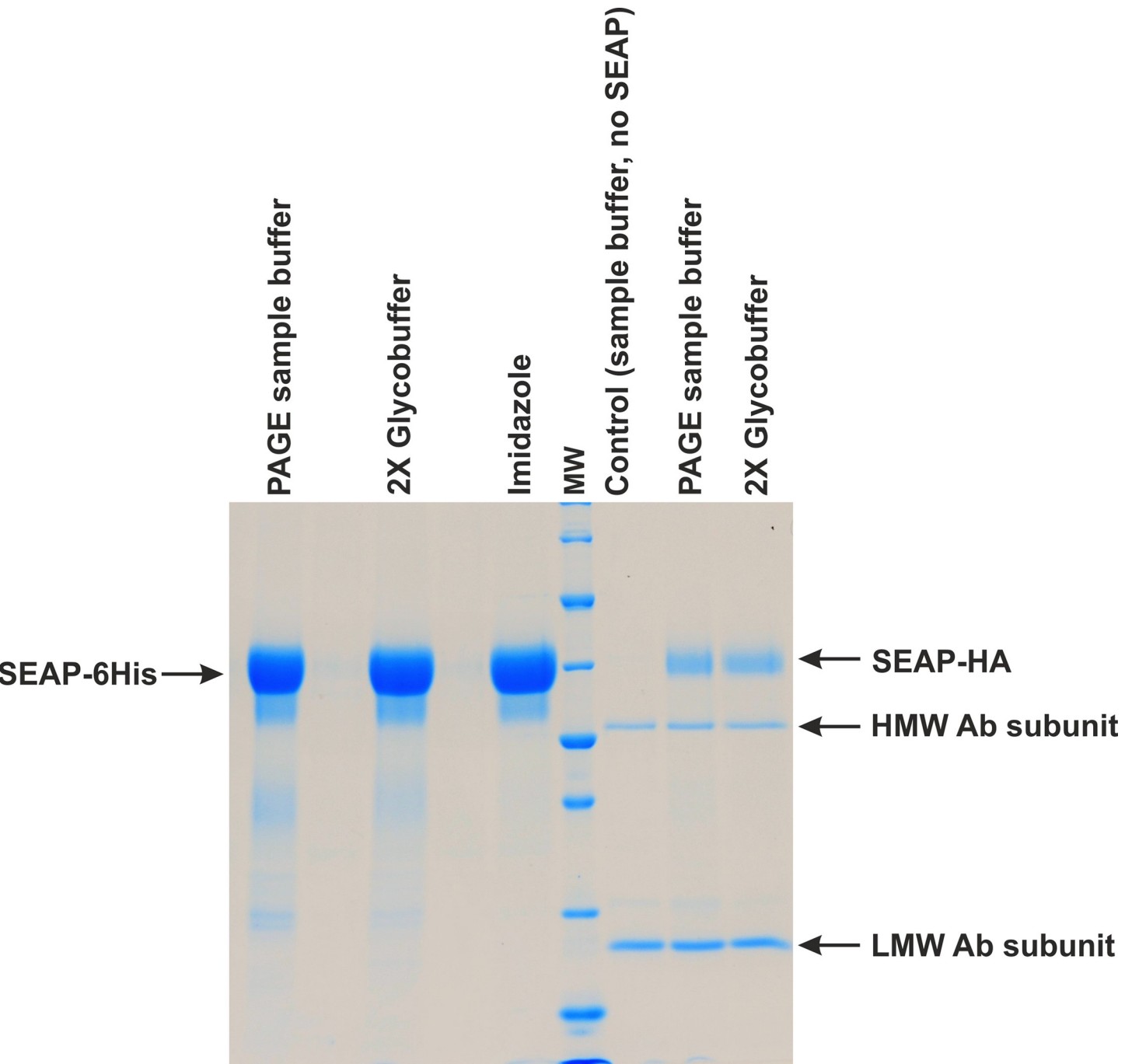

**Fig 1. Purified recombinant SEAP examined by SDS-PAGE analysis.** SEAP was produced in HEK 293 cells and purified from medium containing 10% serum. Lanes 1–3, SEAP-6×His purified on Ni-NTA magnetic agarose beads; lane 4, molecular weight standards; lanes 5–7, SEAP-HA purified on magnetic beads with immobilized anti-HA antibodies. The method of elution is shown at the top of each lane.

get very poor recovery using Ni-NTA column chromatography and batch procedures (typically between 20% and 50%, never more than 70%). There are also additional advantages for using magnetic beads. Firstly, the method allows for low volume elution, and most importantly, it may be done directly with concentrated PNGase denaturing buffer, providing that prolonged incubation at 100˚C for at least 10 minutes (with gentle shaking) is introduced for

optimal recovery. From the recovery point of view, such a procedure is comparable to the commonly used standard elution with imidazole and Laemmli SDS-PAGE buffer, which was also tested (Fig 1, lanes 1–3).

Similar conditions for purification were analyzed for SEAP-HA present in the culture media. Although the secretion rate was comparable to that of SEAP-6×His, the final result was not satisfactory, due to substantial leakage of immobilized anti-HA antibodies from agarose magnetic beads (Fig 1, lanes 4–7). It seems that strong detergents present in elution buffers are not neutral for proteins immobilized on such matrices. Because antibodies are usually glycosylated, it would be not recommended to use this method for *N*-glycosylation analyses.

### 3.3. Introducing new *N*-glycosylation sites by site-directed mutagenesis

In 6×His and HA constructs, we introduced 7 new *N*-glycosylation sites (14 constructs in total were prepared). First, all of them were tested for secretion rate, using QUANTI-Blue reagent (Table 1). The best plasmids selected in regard to the highest secretion rate of transfected cells were used as templates for the second round of site-directed mutagenesis to introduce an

**Table 1. Constructs tested for phosphatase secretion and activity.**

| No. | Construct | Amino acid residues changed to create a new N-glycosylation site (sequon triplet N-X-S/T) | Fusion tag | Phosphatase activity (- = no activity) |
|---|---|---|---|---|
| | | Original SEAP sequences with fusion peptides attached at C-terminus | | |
| 1. | SEAP-His-wt | No | 6xHis | **** |
| 2. | SEAP-HA-wt | No | HA | **** |
| 3. | SEAP-GST-wt | No | GST | * |
| | | Single mutants | | |
| 4. | SEAP-His-36 | $^{38}$Glu → $^{38}$Thr | 6xHis or HA | - |
| 5. | SEAP-His-101 | $^{101}$Ala → $^{101}$Asn | 6xHis or HA | - |
| 6. | SEAP-His-109 | $^{109}$Asp → $^{109}$Thr | 6xHis or HA | * |
| 7. | SEAP-His-152 | $^{152}$Val → $^{152}$Ser | 6xHis or HA | **** |
| 8. | SEAP-His-278 | $^{278}$Gln → $^{278}$Asn | 6xHis or HA | ***** |
| 9. | SEAP-His-477 | $^{493}$Pro → $^{493}$Asn | 6xHis or HA | * |
| 10. | SEAP-His-493 | $^{493}$Pro → $^{493}$Asn | 6xHis or HA | ** |
| | | Double mutants | | |
| 11. | SEAP-His-36-278 | $^{38}$Glu → $^{38}$Thr, $^{278}$Gln → $^{278}$Asn | 6xHis or HA | - |
| 12. | SEAP-His-101-278 | $^{101}$Ala → $^{101}$Asn $^{278}$Gln → $^{278}$Asn | 6xHis or HA | - |
| 13. | SEAP-His-109-278 | $^{109}$Asp → $^{109}$Thr $^{278}$Gln → $^{278}$Asn | 6xHis or HA | ** |
| 14. | SEAP-His-152-278 | $^{152}$Val → $^{152}$Ser, $^{278}$Gln → $^{278}$Asn | 6xHis or HA | ***** |
| 15. | SEAP-His-477-278 | $^{278}$Gln → $^{278}$Asn, $^{477}$Glu → $^{477}$Asn | 6xHis or HA | *** |
| 16. | SEAP-His-493-278 | $^{278}$Gln → $^{278}$Asn, $^{493}$Pro → $^{493}$Asn | 6xHis or HA | *** |

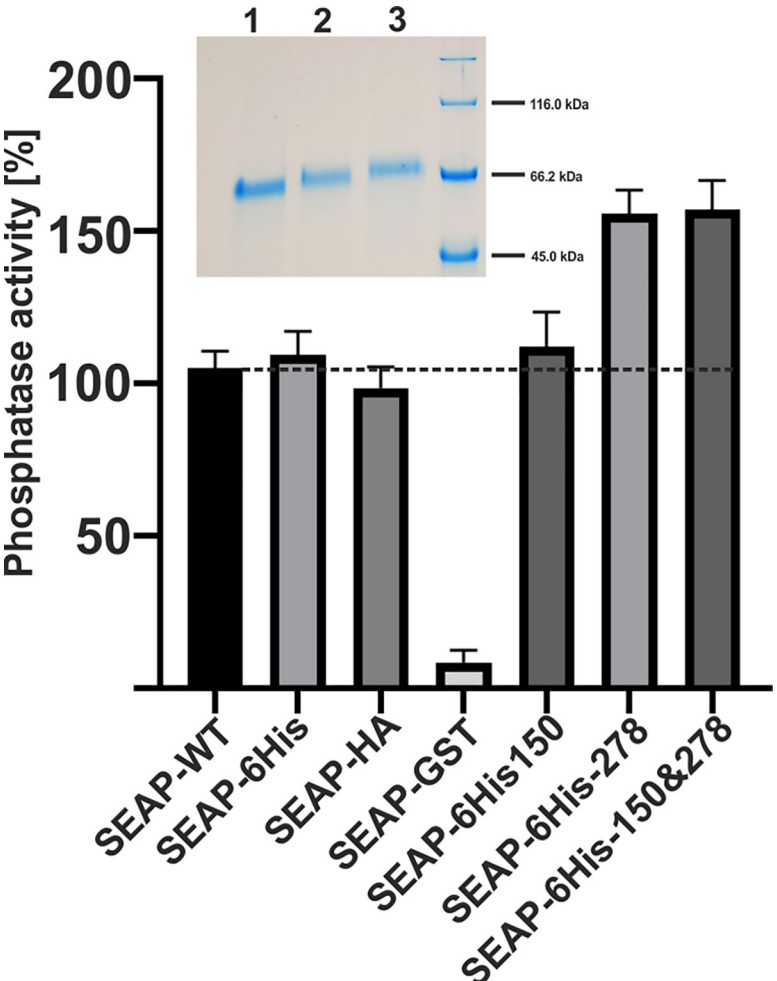

**Fig 2. Relative secretion level of SEAP from HEK293T cells transfected with modified psiTEST vector.** The amount of phosphatase was determined using QUANTI-Blue reagent, as described in Materials and Methods. All experiments were performed in triplicate. All values were expressed as means ± SD. The calculated secretion rate for SEAP-6×His with additional *N*-glycosylation sites introduced at positions 150 and 278 was approximately 12 µg per 1 ml of the medium. The inset at the top presents SDS-PAGE of SEAP-6×His purified on magnetic Ni-NTA beads from serum-free media collected from HEK293T cells, transfected with vector coding wild-type enzyme (lane 1), enzyme with additional glycosylation site at position 278 (lane 2), and double mutant SEAP with glycosylation sites introduced at positions 150 and 278 (lane 3), respectively. 1 µg of the protein was applied into each lane; lane 4 –molecular weight standards. Statistical significance was assigned to p-value < 0.05. ns, not significant; *, p < 0.05; **, p < 0.01; ***, p < 0.001.

additional *N*-glycosylation site (12 additional double mutants in total were prepared). These constructs were tested for phosphatase activity detected in the culture media after transfection. Results from HEK293T cell cultures, shown in Fig 2, confirmed that creation of a new *N*-glycosylation site at position 278 increased secretion of the protein to approximately 150% as compared to secretion of the unmodified, native protein. It seems that there was also a slight improvement when the new *N*-glycosylation site was added at position 150; however, this was not statistically significant. Higher molecular mass of the new protein variants confirmed that the newly introduced glycosylation sites were occupied (top inset in Fig 2). Finally, we concluded that both single SEAP-6×His_278 and double SEAP-6×His_150&278 mutants were comparable in regard to secretion rate, clearly better when compared to the native SEAP. To

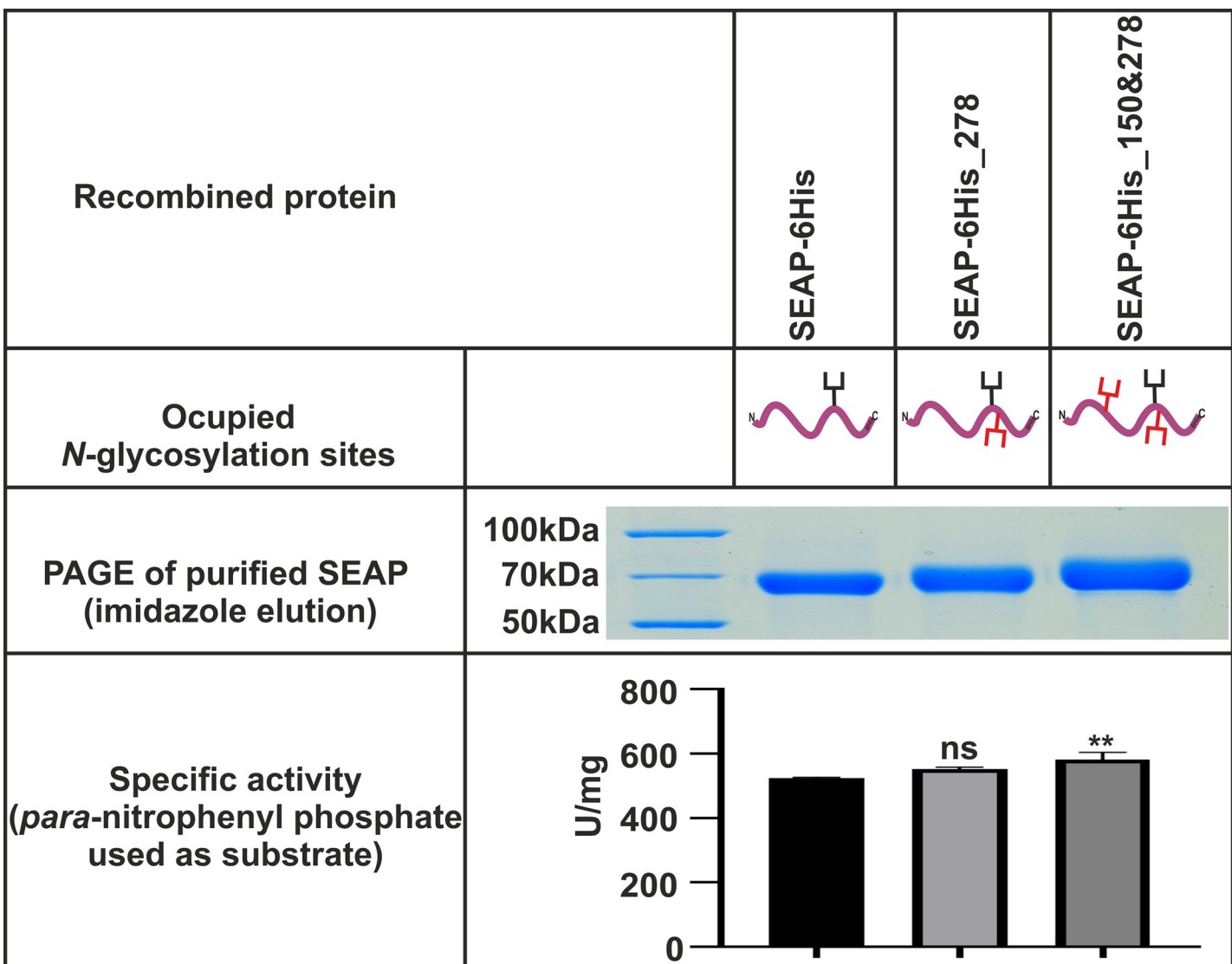

**Fig 3. Comparison of three variants of recombined and enzymatically active SEAP.** The enzyme was bound on magnetic beads and released with imidazole. Specific activity was calculated from end-point reactions of *p*-nitrophenyl phosphate as substrate (see Materials and Methods for details). Statistical significance was assigned to p-value < 0.05. ns, not significant; *, p < 0.05; **, p < 0.01; ***, p < 0.001.

confirm our strategy of SEAP concentration measurements calculated from its activity in the medium, the specific activity of purified variants after mutagenesis was tested on native, non-denatured proteins, gently released from Ni-NTA magnetic beads using 250 mM of imidazole. Here, the homogeneous protein concentration was estimated directly from the enzyme molar coefficient (for details see Materials and Methods). Data presented in Fig 3 show that specific activity of the two best SEAP variants with newly introduced *N*-glycosylation sites were comparable to specific activity of SEAP-6His protein, used as a template for site-directed mutagenesis. The calculated phosphatase specific activity values were 523±3 U/mg, 552±9 U/mg and 581± 19 U/mg, for SEAP-6His, SEAP-278 and SEAP-150&278, respectively.

The effect of the same mutation in the SEAP-HA construct was similar to 6×His plasmids or only slightly less efficient (Table 1). However, we decided to abort the next experiments on

this system and we do not recommend it, because of serious problems with purification (see above, paragraph 3.2).

### 3.4. Glycosylation profiles of SEAP produced and secreted by HEK293T, HepG2 and CHO cells

Fig 4 shows the profiles of total pools of 2-AB labeled *N*-glycans separated on the GlycoSepN amide column. Although it was not our goal to characterize all oligosaccharide structures, it is clear that examined cells can produce SEAP with mature *N*-glycans. It seems that the complexity of glycan pools derived from CHO cells was higher when compared to HEK293T and HepG2 cells. More importantly, we analyzed the high-mannose *N*-glycan content in SEAP produced by HEK293T cells. In contrast to the high rate of glycans with terminally bound alpha-mannoses derived from the cell lysate glycoproteins of HEK293T (more than 80%), in SEAP only about 15% of total glycan content was detected as high-mannose type (Fig 5).

## 4. Final remarks

A schematic view of the new procedure is shown in Fig 6. The newly constructed plasmids, modified by us, work better than those coded for the original SEAP. The addition of the C-terminal 6×His tag seems to be optimal in regard to protein secretion and purification. Introducing a new *N*-glycosylation site at position 278 (and, to lesser extent, at position 150) increases secretion of the enzyme to the cell culture medium. SEAP-6×His_150 and SEAP-6×His_278 are glycosylated mostly with complex *N*-glycans, whereas the percentage of high-mannose structures is low. We recommend using Ni-NTA agarose magnetic beads for isolation of the recombinant SEAP not only because of the high recovery rate, but also because of the possibility of elution in low volumes, if using 2× concentrated glycoprotein denaturing buffer. Such elution mixtures may be directly used for the PNGase deglycosylation procedure. Finally, we estimated the concentration of secreted protein in the media. Although it was not our intention to test the optimal conditions for cells during SEAP secretion and the protein level was strongly dependent on the physiological state of the cells, way of transfections, time of media collection, confluency of the cells, etc., the typical concentration of modified SEAP in the media was estimated in the range 50–150 μg/ml for HEK293T, 20–100 μg/ml for HepG2 and 10–50 μg/ml for CHO cells, respectively. Most importantly, modified SEAP with two additional and occupied glycosylation sites (at positions 150 and 278) is three times more highly glycosylated than the initial enzyme. Assuming high recovery of SEAP released from magnetic agarose beads, such a level of expression enables glycosylation profiles to be analyzed (for example using mass spectrometry or exoglycosidase sequencing) in all examined cell lines, even from as low as 0.5–2 ml of the standard cell culture medium. The procedure may be also scaled up without decrease of the final efficiency.

It must also be noted that there are some disadvantages of the presented method; however, these are common to all methods based on application of any reporter, recombined protein. In every experiment, the efficiency of mammalian cell transfection must be tested, first of all to choose the best transfection reagent, optimal conditions during gene transfer, and time elapsed for media harvesting. This is usually dependent on the type of used cell line and may vary significantly.

Finally, in our opinion, taking into account the existence of three occupied *N*-glycosylation sites, in contrast to the single *N*-glycosylation site present in the wild-type SEAP, considerably higher secretion rate and improved purification protocol, at least 4–5 times better *N*-glycan recovery might be expected, when compared to methods based on reporter alkaline phosphatase, used to date.

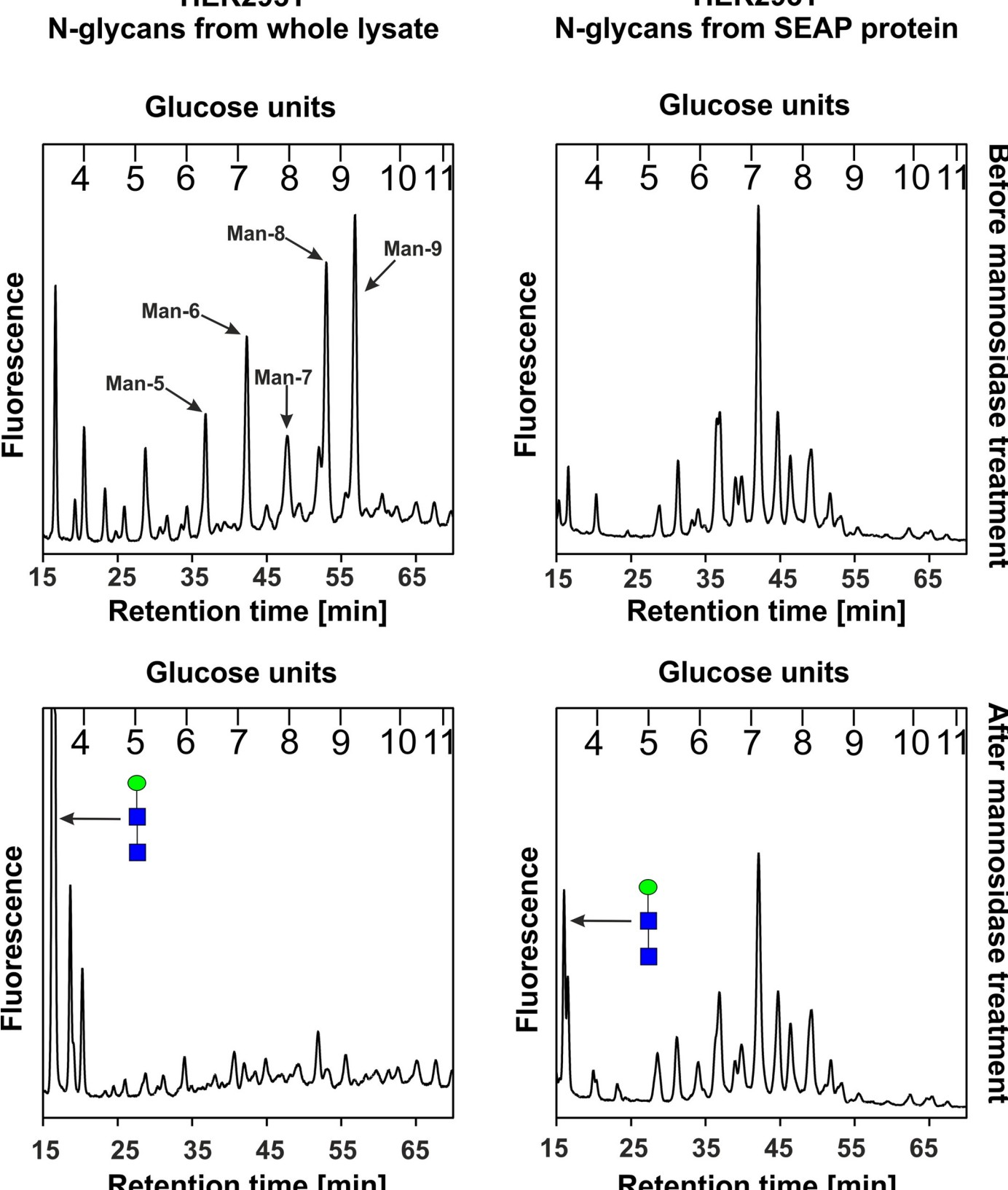

**Fig 4. Analysis of abundance of high-mannose structures.** *N*-glycan profiles of HEK293T total lysate and SEAP-6×His with new glycosylation sites at 150 and 278 positions, purified on Ni-NTA beads from standard medium, are shown. Top panels–before mannosidase treatment; bottom panels–after mannosidase digestion. Peak on the left shows the product of digestion of all high-mannose type structures (blue, solid square–*N*-acetylglucosamine, green, solid circle–mannose).

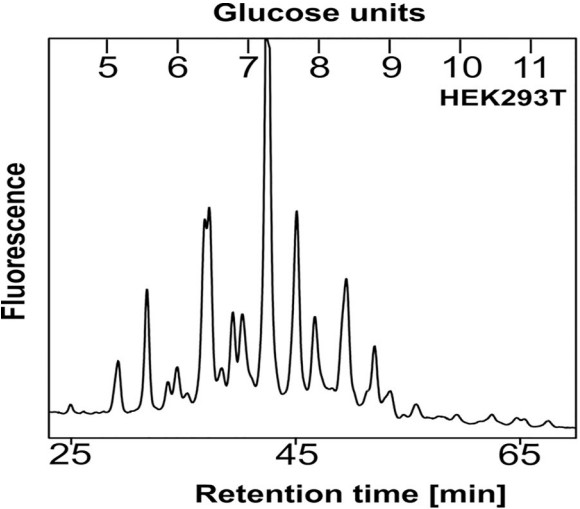

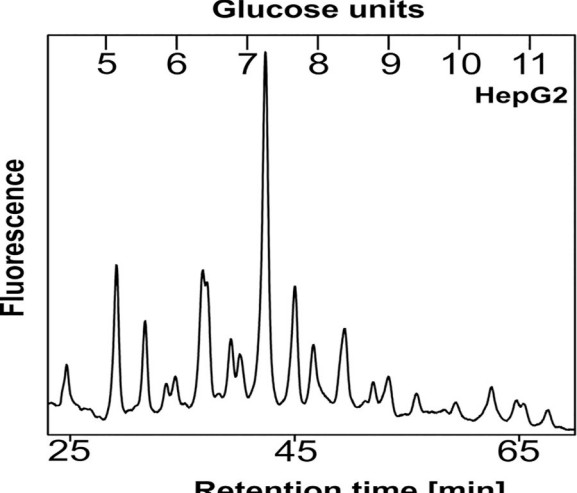

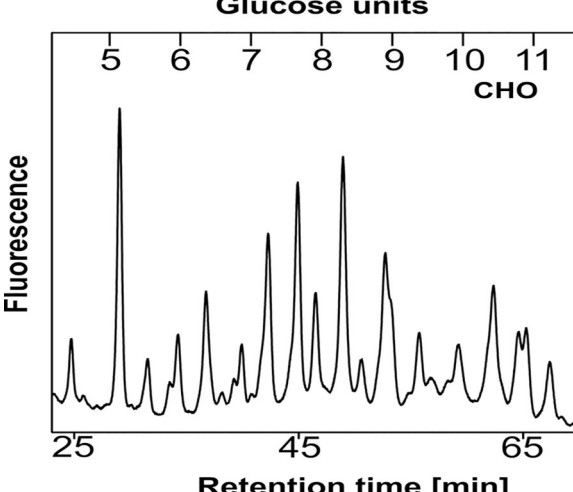

**Fig 5.** *N*-glycosylation profiles of modified SEAP, purified from media of HEK293T, HepG2 and CHO cell cultures.

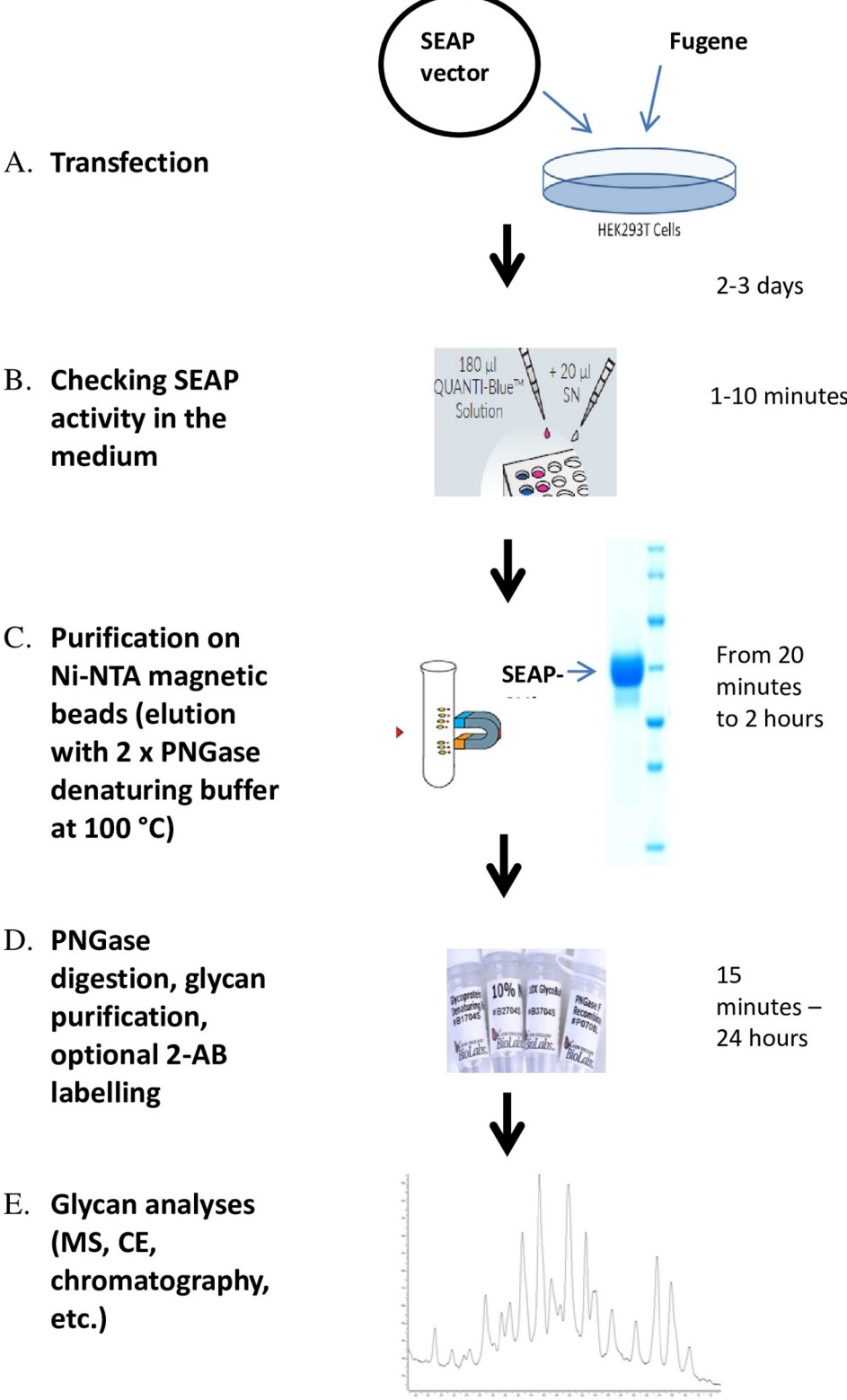

**A. Transfection**

2-3 days

**B. Checking SEAP activity in the medium**

1-10 minutes

**C. Purification on Ni-NTA magnetic beads (elution with 2 x PNGase denaturing buffer at 100 °C)**

From 20 minutes to 2 hours

**D. PNGase digestion, glycan purification, optional 2-AB labelling**

15 minutes – 24 hours

**E. Glycan analyses (MS, CE, chromatography, etc.)**

2-AB N-glycan profile of SEAP purified from 2 ml of DMEM, 3 days after transfection of HEK293T cells

**Fig 6. Schematic description of proposed conditions for *N*-glycan isolation from recombined SEAP.**

## Supporting information

**S1 Fig.**
(PDF)

**S2 Fig.**
(PDF)

**S1 Table.**
(PDF)

**S1 Raw images.**
(PDF)

## Author Contributions

**Conceptualization:** Mariusz Olczak.

**Data curation:** Mariusz Olczak, Bożena Szulc.

**Formal analysis:** Mariusz Olczak.

**Funding acquisition:** Mariusz Olczak.

**Investigation:** Mariusz Olczak, Bożena Szulc.

**Methodology:** Mariusz Olczak.

**Project administration:** Mariusz Olczak.

**Supervision:** Mariusz Olczak.

**Validation:** Bożena Szulc.

**Visualization:** Mariusz Olczak, Bożena Szulc.

**Writing – original draft:** Mariusz Olczak.

**Writing – review & editing:** Mariusz Olczak, Bożena Szulc.

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
