## [Decision Letter · Decision Letter 0]

9 Feb 2021

PONE-D-21-00326

Modified secreted alkaline phosphatase - improved reporter protein for N‑glycosylation analysis

PLOS ONE

Dear Dr. Olczak,

Thank you for submitting your manuscript to PLOS ONE. After careful consideration, we feel that it has merit but does not fully meet PLOS ONE’s publication criteria as it currently stands. Therefore, we invite you to submit a revised version of the manuscript that addresses the points raised during the review process.

We look forward to receiving your revised manuscript.

Kind regards,

Nazmul Haque

Academic Editor

PLOS ONE

Journal Requirements:

Reviewers' comments:

Reviewer's Responses to Questions

**Comments to the Author**

1. Is the manuscript technically sound, and do the data support the conclusions?

Reviewer #1: No

Reviewer #2: Yes

2. Has the statistical analysis been performed appropriately and rigorously? 

Reviewer #1: No

Reviewer #2: N/A

3. Have the authors made all data underlying the findings in their manuscript fully available?

Reviewer #1: No

Reviewer #2: Yes

4. Is the manuscript presented in an intelligible fashion and written in standard English?

Reviewer #1: No

Reviewer #2: Yes

5. Review Comments to the Author

Reviewer #1: This article describes a modified alkaline phosphatase as secreted glycoprotein model. The alkaline phosphatase has been cloned into different mammalian expression plasmids and its expression/secretion into the medium was documented concerning N-Glycosylation driven by sites directed mutagenesis. Different expression systems, expression plasmids, and different purification columns were used.

Here are comments and major concerns about this article:

The title of this article is miss leading, this reviewer is concerned about the suitability of the title to the article contents “Modified secreted alkaline phosphatase - improved reporter protein for N-glycosylation analysis” how the secreted alkaline phosphatase do improve N-glycosylation analysis or vice versa?

What were the hypothesis and aims of the current research?

The methods are not adequate for the specific results and conclusions!

The results are not in line with the conclusions of the article!

This reviewer doesn't see any relation between the glycosylation of alkaline phosphatase and its expression/activity as an enzyme unless the claims are tested one by one with proper methodology! If its expression, then the quantity of the enzyme expression and secretion in the medium should be tested with specific antibodies or other quantitation methods. If its activity, then, an equal amount of the enzyme and control one should be tested in the same assay.

What is the purpose of alkaline phosphatase as a reporter glycoprotein?

Alkaline phosphatase as a reporter enzyme how does this improve glycosylation analysis? The fact that alkaline phosphatase is a reporter enzyme and secreted into the cell culture medium makes it a single case in which its purification and analysis of its N-glycosylation are straightforward? So why is that so important?

The authors describe several expression systems such as HEK293, CHO, and HepJ cells indicating that these three expression systems are capable to express SEAP in different amounts! What is the purpose of this comparative data, how these data contribute to the article? And to the article aims if there are specific aims! How Figure 4 that depicts the N-glycosylation profile of the different expression system contributes to the article?

One of the authors' conclusions was “Additional N-glycosylation sites introduced by

site-directed mutagenesis significantly increased secretion of the protein”

Based on figure 2, this conclusion is incorrect! Figure 2 title is “Relative secretion level of SEAP from HEK293T cells transfected with modified psiTEST vector”. While on the Y-axis of figure 2 alkaline phosphatase activity is depicted, maybe additional glycosylation sites improved the activity but not the secretion of the enzyme, did the authors tested this claim? And how about the significance of the differences in activity between the site-directed mutagenesis and the wt enzyme? Is there any statistical analysis?

Based on the data presented in figure 3, the authors concluded that the total cell lysate glycoproteins contain mannose immature glycoforms, but N-glycoforms of SEAP are mature types. Is that novel data? Is that surprising? How this specific work improves N-glycosylation analysis of secreted proteins while there is no additional information on the N-glycosylation types of SEAP?

Minor comments:

This article needs English editing

The introduction should be rewritten. The first two sentences of the introduction are saying the same!

Introduction paragraph 4, “4) the reporter protein possesses at least three glycosylation sites. (the original SEAP contains two N-glycosylation sites, from which only one is occupied [15,16]), which usually makes N-glycan profile more complex”.

What do you mean in this expressions? If only one glycosylation site is occupied does that make N-glycan profile more complex?

Results and discussion:

The first paragraph “It seems that relatively big size of GST (25 kDa) would

be a negative factor for secretion of the enzyme.”

The authors should be careful by concluding that the reason for low secretion is that the protein size is big!

3.3 “In 6×His and HA constructs, we introduced 7 new N-glycosylation sites (14 constructs in total were prepared). First, all of them have been tested for secretion rate, using QUANTI-Blue reagent (Supplementary Table 1). The best plasmids were used as templates for the second round of site-directed mutagenesis to introduce additional N-glycosylation site (additional 12 double mutants in total were prepared).”

Rewrite please this is unclear, how many sites you have mutated in each construct?

3.3 the second paragraph “The effect of the same mutation in SEAP-HA construct was similar to 6×His plasmids or only slightly less efficient”

What do you mean by only slightly less efficient? Why it is important to state that?

The last three lines in results and discussion “In contrast to high rate of glycans with terminally bound alphamannoses derived from the cell lysate glycoproteins of HEK293T (more than 80%), in SEAP only about 15% of total glycan content was detected as high-mannose type.

Where this data come from?

Reviewer #2: This is a well written manuscript describing a technique to measure glycosylation using secreted alkaline phosphatase modified to bear glycosylation sites inanition to the preexisting sites. Detailed methodology and conditions described will enable other researchers to use this technique. Overall it is an excellent paper. However I have the following comments.

1. The legends for the figures are very brief making it difficult for someone who is not a glycobiologist to understand what the figures mean. Ex Fig 3 and 4. What do the axis mean and what is interpreted from the figure should be better described in the legend and better expanded in the text.

2. One aspect that is confusing for this reviewer is whether glcolsylation sequence in the additional sites are the same as the glycosylation of the pre existing site. This could have been done with the protein modified to bear one additional site and compared to the preexisting site. I am assuming that the glycosylation sequence should be known for the preexisting sites. I am not sure what you gain from knowing glycosylation in all the new sites if we do not know if the new site glycosylation has no relation to glycosyl residue in the preexisting site.

3. It would have been interesting to choose to cell lines one from human and one from non human primate for comparison.

4.What and how exactly the modified protein can be used could have been described in the discussion.

6. PLOS authors have the option to publish the peer review history of their article (what does this mean?). If published, this will include your full peer review and any attached files.

Reviewer #1: No

Reviewer #2: No

---

## [Author Response · Author response to Decision Letter 0]

4 Apr 2021

Journal Requirements:

 The manuscript was re-formatted according to PLOS ONE requirements

The original gel images are downloaded as Supporting Information

Phrase "data not shown" was removed from the new version of the manuscript

Information about attached Supplementary files are placed at the end of our manuscript

5. Review Comments to the Author

Reviewer #1: 

Thank you very much for the critical review of our manuscript. We agree with the majority (but not all) received comments. Looking at the comments, we understood that two serious problems with our work were pointed. First comment is connected with the way of investigation of protein secretion level, which was directly calculated from SEAP enzymatic activity. The second problem concerns the aim of this study. We tried to overcame those problems with newly introduced experiments and also by re-writing the text of the manuscript to be available for broader audience, for readers not only specialized in glycobiology. Below please find our detailed explanations for each comment expressed to the 1st version of our manuscript: 

This article describes a modified alkaline phosphatase as secreted glycoprotein model. The alkaline phosphatase has been cloned into different mammalian expression plasmids and its expression/secretion into the medium was documented concerning N-Glycosylation driven by sites directed mutagenesis. Different expression systems, expression plasmids, and different purification columns were used.

Here are comments and major concerns about this article:

The title of this article is miss leading, this reviewer is concerned about the suitability of the title to the article contents “Modified secreted alkaline phosphatase - improved reporter protein for N-glycosylation analysis” how the secreted alkaline phosphatase do improve N-glycosylation analysis or vice versa?

We agree, the Title would be missleading. We changed it.

What were the hypothesis and aims of the current research?

 Our work described genetic modification of plasmid (vector) for using in N-glycosylation analyses, the major type of protein post-translational modifications. The changed psiTEST plasmid was examined for its functionality in futher structural studies on N-glycans, sugar oligomers which are covalently attached to the recombined SEAP (secretion, occupancy of introduced sequons and improvements in purification from culture media were tested). We feel that researches not familiar with glycobiology are not aware with serious technical problems related to glycan analyses in various cells and tissues; it is well known that glycobiogists usually struggle to isolate representative pools of glycoconiugates. Glycosylation belongs to very common post-translational modifications, however, many important questions and problems, especially glycotransferase regulation, are still not known. Many strategies were proposed to overcame technical problems to possess reliable procedures for structural analysis of glycans, knowledge essential to investigate cellular glycosylation network. The choice of reporter, secretable glycoprotein in genetically modified cell line as a source of glycans was introduced many times in the past. In our oppinion, the manuscript presents one of the best system for such analyses, much better than previously proposed. In the Introduction chapter we add the new paragraph which explains all these problems. We hope this would help to understand the advantages of using such reporter glycoprotein for glycoanalyses.

The methods are not adequate for the specific results and conclusions!

The results are not in line with the conclusions of the article!

We hope that after introduced changes and additional experiments (see Figure 3), the conclusions, supplemented with additional comments match the results. From our point of view, now methods are also adequate for all presented results.

This reviewer doesn't see any relation between the glycosylation of alkaline phosphatase and its expression/activity as an enzyme unless the claims are tested one by one with proper methodology! If its expression, then the quantity of the enzyme expression and secretion in the medium should be tested with specific antibodies or other quantitation methods. If its activity, then, an equal amount of the enzyme and control one should be tested in the same assay.

We agree with the reviewer that the level of secretion must be analysed with more appropriate method. If the enzyme activity is used, all protein variants must express the same (or very similar) activity at standard conditions. We decided to analyse specific activity of three variants, examined in secretion experiments (see Materials and methods and Figure 3). Additionally the calculated molar absorbance coefficient, which was used to establish the concentration of homogenous SEAP for kinetic studies. Although we observed some, tiny differences between specific activities of SEAP variants, it is clear that this can not exclude enzyme activity measurments for estimation of the SEAP concentration in the culture medium.

What is the purpose of alkaline phosphatase as a reporter glycoprotein?

Alkaline phosphatase as a reporter enzyme how does this improve glycosylation analysis? The fact that alkaline phosphatase is a reporter enzyme and secreted into the cell culture medium makes it a single case in which its purification and analysis of its N-glycosylation are straightforward? So why is that so important?

We changed the Introduction section to explain why such methods are important, first of all as a real and representative source of mature N-glycans intended for future structural studies. We also proposed the optimal method of SEAP isolation, faster, efficient and ready for enzymatic de-glycosylation. The text of the manucript was supplemented with new paragraphs to be understable for broader audience.

The authors describe several expression systems such as HEK293, CHO, and HepJ cells indicating that these three expression systems are capable to express SEAP in different amounts! What is the purpose of this comparative data, how these data contribute to the article? And to the article aims if there are specific aims! How Figure 4 that depicts the N-glycosylation profile of the different expression system contributes to the article?

Here, we do not understand this comment. We feel that the reviewer did not know, that the three mentioned cell lines are frequently used in glycobiology. Especially CHO cell line, derived from Chinese Hamster ovary many years ago, looks as no.1 system for glycoanalyses, first of all because of avaliability of many glycomutants (for example Lec-type cell lines generated by dr. Pamela Stanley from late 80-ties last century to present), with disturbed glycosylation pathways. These cell lines were not accidentally choosen. Authors used to work with them for many years, and characterized them especially to analyse nucleotide sugar transporter action, the process which delivers substrates for glycosylation in ER and Golgi. But these profillings (now presented on figures 4 and 5) must be perfomed, for example to show, that the new genetic construct is not toxic for the tested cell lines and also to analyse the quality of isolated N-glycans, first of all, if the non-mature fraction is not dominating over mature N-glycans. Finally, our goal was not to compare the glycoprofiles but to show that the FULL procedure, from transfection to enzymatic de-glycosylation (see schematic figure 6) works well for all investigated lines. And even for the worse recorded secretion rate, it is clear that it may be apllied with success to get purified pool of N-glycans, detectable in chromatographic techniques after fluorescent labelling, ready for the next studies. 

One of the authors' conclusions was “Additional N-glycosylation sites introduced by

site-directed mutagenesis significantly increased secretion of the protein”

Based on figure 2, this conclusion is incorrect! Figure 2 title is “Relative secretion level of SEAP from HEK293T cells transfected with modified psiTEST vector”. While on the Y-axis of figure 2 alkaline phosphatase activity is depicted, maybe additional glycosylation sites improved the activity but not the secretion of the enzyme, did the authors tested this claim? And how about the significance of the differences in activity between the site-directed mutagenesis and the wt enzyme? Is there any statistical analysis?

We add additional experiments to show similar phoshatase activity of all variants. Statistics was present in the previous version, now the manuscript is supplemented with additional information in the Materials and methods and also in the captions to figures 2 and 3.

Based on the data presented in figure 3, the authors concluded that the total cell lysate glycoproteins contain mannose immature glycoforms, but N-glycoforms of SEAP are mature types. Is that novel data? Is that surprising? How this specific work improves N-glycosylation analysis of secreted proteins while there is no additional information on the N-glycosylation types of SEAP?

Again, we fell that the reviewer is not familiar with glycobiology and glycotechniques. Yes, indeed, it is well known that the high level of high mannose type glycans is typical for many mammalian cell lines. For that reason researches are searching for other ways to analyse mature forms only (some examples of such strategies were explained in the Introduction). However, we undestand that readers of Plos One may be also puzzled and the objections from the reviewer may be admissible. Because of that, although there were some data about this phenomenon (domination of mannose structures) in the 1st version of the text, we decided to supplement the Introduction chapter to explain the necessity of looking for new, improved methods for isolation of mature glycans only.

Minor comments:

This article needs English editing

The introduction should be rewritten. The first two sentences of the introduction are saying the same!

We agree with the reviewer. The sentence was removed from the text. Introduction was rewritten to be more understable for general audience. We also made some minor language improvments in the text.

Introduction paragraph 4, “4) the reporter protein possesses at least three glycosylation sites. (the original SEAP contains two N-glycosylation sites, from which only one is occupied [15,16]), which usually makes N-glycan profile more complex”. What do you mean in this expressions? If only one glycosylation site is occupied does that make N-glycan profile more complex?

We agree with the comments, the sentence was modified.

Results and discussion:

The first paragraph “It seems that relatively big size of GST (25 kDa) would

be a negative factor for secretion of the enzyme.”

The authors should be careful by concluding that the reason for low secretion is that the protein size is big!

We agree, the sentence was removed. The reason of low activity/secretion of GST-tagged SEAP is not proved.

3.3 “In 6×His and HA constructs, we introduced 7 new N-glycosylation sites (14 constructs in total were prepared). First, all of them have been tested for secretion rate, using QUANTI-Blue reagent (Supplementary Table 1). The best plasmids were used as templates for the second round of site-directed mutagenesis to introduce additional N-glycosylation site (additional 12 double mutants in total were prepared).”

Rewrite please this is unclear, how many sites you have mutated in each construct?

For clarity, we transfer Table 1 from Supplementary material to the main text of the manuscript. In this table all tested constructs (including those from preliminary studies) are listed. We also made some changes in mentioned fragment of the manuscript.

3.3 the second paragraph “The effect of the same mutation in SEAP-HA construct was similar to 6×His plasmids or only slightly less efficient”

What do you mean by only slightly less efficient? Why it is important to state that?

These results of our preliminary experiments are summarised in Table 1, now the part of the main text. From our point of view the cited statement is important, because does not exclude HA tagged SEAP as a potential tool in future studies, for example to analyse secretion mashinery of selected cell lines, with the use of very specific, anti-HA antibodies, easily available on the market (this is impossible for 6xHis tag, due to low specificity of anti-6xHis antibodies). We struggled with purification of HA-tagged SEAP and stopped experiments on HA-SEAP as a source of glycans, however, we think that HA-SEAP still may be used in some experiments, if purification on immobilised HA-antibodies is avoided.

The last three lines in results and discussion “In contrast to high rate of glycans with terminally bound alphamannoses derived from the cell lysate glycoproteins of HEK293T (more than 80%), in SEAP only about 15% of total glycan content was detected as high-mannose type.

Where this data come from?

We included additional explanations in the Results an discussion part. In Materials and methods the phrase explaining the way of glycan pools calculations from fluorescent peak areas were added. 

Reviewer #2: This is a well written manuscript describing a technique to measure glycosylation using secreted alkaline phosphatase modified to bear glycosylation sites inanition to the preexisting sites. Detailed methodology and conditions described will enable other researchers to use this technique. Overall it is an excellent paper. However I have the following comments.

Thank you very much for the critical review of our manuscript. We agree with the majority (but not all) received comments. Below please find the detailed anwers to the reviewer's concerns:

1. The legends for the figures are very brief making it difficult for someone who is not a glycobiologist to understand what the figures mean. Ex Fig 3 and 4. What do the axis mean and what is interpreted from the figure should be better described in the legend and better expanded in the text.

Description for Figures (now numbered as Figure 4 and 5) were changed to be more understable for general audience

2. One aspect that is confusing for this reviewer is whether glcolsylation sequence in the additional sites are the same as the glycosylation of the pre existing site. This could have been done with the protein modified to bear one additional site and compared to the preexisting site. I am assuming that the glycosylation sequence should be known for the preexisting sites. I am not sure what you gain from knowing glycosylation in all the new sites if we do not know if the new site glycosylation has no relation to glycosyl residue in the preexisting site.

We agree with the comment but this was not our goal to analyse all occupied N-glycosylation sites in details. The aim of this study was to improve the glycoprotein (SEAP, to be produced in at higher level, was easily purified in relatively short time, and be available for researches working in laboratories with standard equipment). We proved, that the final product contained 3 occupied glycosylation sites (see at higher molecular weight of constructs 278 and 150&278 visibleon figure 2 and figure 3) and was secreted into the medium very efficiently, more than initial SEAP. 

3. It would have been interesting to choose to cell lines one from human and one from non human primate for comparison. 

We agree with the reviewer, but our goal was not to analyse glycoprofiles of many cell lines. We use some cells to show functionality of the newly designed genetic vector. And we analysed two human cell lines and one, derived from Chinese Hamster ovary (CHO). These lines are typically used by researches for glycoanalyses. However, we are sure that the system may work for broad spectrum of mammalian cell lines, including non human primate. Assuming the high efficiency of transfection - we add some comments about that at the end od Results and discussion part.

4.What and how exactly the modified protein can be used could have been described in the discussion.

We agree that the previous version of the manuscript was very compact and might be not easy to understand by researches, not specialized in glycobiology. We added sugested paragraph at the end of Results and discussion section.

---

## [Decision Letter · Decision Letter 1]

26 Apr 2021

PONE-D-21-00326R1

Modified secreted alkaline phosphatase as improved reporter protein for N‑glycosylation analysis

PLOS ONE

Dear Dr. Olczak,

Thank you for submitting your manuscript to PLOS ONE. After careful consideration, we feel that it has merit but does not fully meet PLOS ONE’s publication criteria as it currently stands. Therefore, we invite you to submit a revised version of the manuscript that addresses the points raised during the review process.

We look forward to receiving your revised manuscript.

Kind regards,

Nazmul Haque

Academic Editor

PLOS ONE

Journal Requirements:

Additional Editor Comments (if provided):

Following revision the manuscript has been improved significantly. However, this manuscript requires significant English editing to improve its readability.

Reviewers' comments:

Reviewer's Responses to Questions

**Comments to the Author**

1. If the authors have adequately addressed your comments raised in a previous round of review and you feel that this manuscript is now acceptable for publication, you may indicate that here to bypass the “Comments to the Author” section, enter your conflict of interest statement in the “Confidential to Editor” section, and submit your "Accept" recommendation.

Reviewer #1: All comments have been addressed

Reviewer #2: All comments have been addressed

2. Is the manuscript technically sound, and do the data support the conclusions?

Reviewer #1: Yes

Reviewer #2: Yes

3. Has the statistical analysis been performed appropriately and rigorously? 

Reviewer #1: Yes

Reviewer #2: Yes

4. Have the authors made all data underlying the findings in their manuscript fully available?

Reviewer #1: Yes

Reviewer #2: Yes

5. Is the manuscript presented in an intelligible fashion and written in standard English?

Reviewer #1: No

Reviewer #2: Yes

6. Review Comments to the Author

Reviewer #1: (No Response)

Reviewer #2: The revised manuscript addresses all the comments and queries raised by me.

The authors have also included new data.

Along with detailed response to the other reviewer the manuscript is much improved.

I have no reservations or comments for the revised manuscript

7. PLOS authors have the option to publish the peer review history of their article (what does this mean?). If published, this will include your full peer review and any attached files.

Reviewer #1: No

Reviewer #2: No

---

## [Author Response · Author response to Decision Letter 1]

1 May 2021

Reviever 1

Thank you very much for your comments.

The manuscript was checked and improved by native English speaker. 

Reviewer 2

No comments, the reviewer accepted the manuscript after 1st revision.

---

## [Editor Report · Decision Letter 2]

4 May 2021

Modified secreted alkaline phosphatase as an improved reporter protein for N‑glycosylation analysis

PONE-D-21-00326R2

Dear Dr. Olczak,

We’re pleased to inform you that your manuscript has been judged scientifically suitable for publication and will be formally accepted for publication once it meets all outstanding technical requirements.

Kind regards,

Nazmul Haque

Academic Editor

PLOS ONE
---

## [Editor Report · Acceptance letter]

14 May 2021

PONE-D-21-00326R2 

Modified secreted alkaline phosphatase as an improved reporter protein for *N*‑glycosylation analysis 

Dear Dr. Olczak:

I'm pleased to inform you that your manuscript has been deemed suitable for publication in PLOS ONE. Congratulations! Your manuscript is now with our production department. 

Kind regards, 

on behalf of

Dr. Nazmul Haque 

Academic Editor

PLOS ONE